# Soft Colloidal Particles at Fluid Interfaces

**DOI:** 10.3390/polym14061133

**Published:** 2022-03-11

**Authors:** Eduardo Guzmán, Armando Maestro

**Affiliations:** 1Departamento de Química Física, Facultad de Ciencias Químicas, Universidad Complutense de Madrid, Ciudad Universitaria s/n, 28040 Madrid, Spain; 2Instituto Pluridisciplinar, Universidad Complutense de Madrid, Paseo de Juan XXIII, 28040 Madrid, Spain; 3Centro de Física de Materiales (CSIC, UPV/EHU), Paseo Manuel de Lardizabal 5, 20018 San Sebastian, Spain; 4IKERBASQUE—Basque Foundation for Science, Plaza Euskadi 5, 48009 Bilbao, Spain

**Keywords:** deformability, interfaces, microgels, softness, interfacial tension, reconfigurable

## Abstract

The assembly of soft colloidal particles at fluid interfaces is reviewed in the present paper, with emphasis on the particular case of microgels formed by cross-linked polymer networks. The dual polymer/colloid character as well as the stimulus responsiveness of microgel particles pose a challenge in their experimental characterization and theoretical description when adsorbed to fluid interfaces. This has led to a controversial and, in some cases, contradictory picture that cannot be rationalized by considering microgels as simple colloids. Therefore, it is necessary to take into consideration the microgel polymer/colloid duality for a physically reliable description of the behavior of the microgel-laden interface. In fact, different aspects related to the above-mentioned duality control the organization of microgels at the fluid interface, and the properties and responsiveness of the obtained microgel-laden interfaces. This works present a critical revision of different physicochemical aspects involving the behavior of individual microgels confined at fluid interfaces, as well as the collective behaviors emerging in dense microgel assemblies.

## 1. Introduction

The assembly of colloidal particles at fluid interfaces is currently exploited in a broad range of industrial and technological fields, ranging from food, oil, and cosmetic industries to pharmaceutical formulations, and from the manufacturing of advanced coatings to biomedical applications [1,2,3,4,5,6,7,8]. In particular, the use of soft colloidal particles gained importance in the above applications in recent years [9,10].

Soft colloidal particles can be fabricated by cross-linked polymer networks of colloidal dimensions swollen by a good solvent, commonly denominated as microgels. They are characterized by a polymer/colloid duality. This can be understood by considering the chemistry and morphology of microgels. Microgels are soft particles characterized by a fuzzy structure, which presents dangling chains in their swollen state, whereas a behavior reminiscent to that expected for hard colloids emerges upon their shrinking. This leads to a very different organization of the polymer network depending on the particle hydration. Swollen microgels presents an open structure characterized by their diffuse outer boundary, and the strong solvation of its inner segments, which favors the mobility of solvents and solute molecules within a network, and their expansion until the limit imposed by the chain elasticity. This contributes to the ability of microgels to undergo reversible network deformations by the uptake or expulsion of large quantities of solvent (good solvent conditions) upon the application of external stimuli, which emerges as one of the most outstanding properties of this type of particle [11,12,13,14,15,16]. This volume phase transition (VPT), commonly triggered by external perturbations (interfacial tension forces or external physical stimuli), provides a broad range of novel structure/function relationships to microgel-laden interfaces [17,18].

Solvated microgel networks can be compressed upon adsorption to fluid interfaces, leading to the deformation of the particle shape, to adapt their conformation to the interface. These morphological changes also control the dynamic behavior and interfacial packing of microgels at the interface, which, in turn, dominate the interfacial tension and rheological properties of the microgel-laden interfaces [19], playing a very important role in emulsion and foam stabilization [20,21]. In particular, the stretching of soft particles observed when assembled at fluid interfaces provides a better stability of the interface than the adsorption of traditional hard colloids. This can be rationalized by the higher adsorption energy of microgels (several orders of magnitude than that of hard particles with similar dimensions) [22]. Furthermore, the bulk properties of microgels favor their adsorption when they are close to fluid interfaces without fulfilling the amphiphilicity requirements [23].

The most common microgel studied in the literature, to date, is the one built using poly(N-isopropyl acrylamide) (PNIPAM). This polymer allows for the preparation of thermosensitive microgels presenting the ability to undergo reversible temperature-triggered swelling/shrinking transitions around physiological human temperatures [24,25,26]. The properties of this type of gel can be easily tuned during the synthesis process by including co-monomers, e.g., acrylic or methacrylic acid, which enables a pH-triggered change of the microgel ionization degree, and hence the microgel swelling degree [27]. Additionally, the fabrication of microgel particles based on proteins, e.g., whey protein or β-lactoglobulin, gained interest in recent years due to their potential application in different food products [18,20].

Given the above considerations, despite the broad interest in the adsorption of microgels to fluid/fluid interfaces, the current knowledge about the effect of the specific characteristics of the microgels on their interfacial organization and mechanical properties is scarce, even though it emerges as a critical point on the design of reconfigurable interfaces for specific applications [21]. This review provides an updated perspective of the current knowledge about the assembly of microgel layers at fluid/fluid interfaces, and how different physico-chemical aspects can be exploited to obtain reconfigurable soft interfaces for different technological purposes, paying special interest to their uses for emulsion stabilization.

## 2. Understanding the Adsorption of Soft Particles-to-Fluid Interfaces

The dual character of microgels plays a central role in their ability to be adsorbed to the fluid interfaces [11,28]. Microgels are surface-active particles that adsorb to fluid interfaces reducing the interfacial tension between two immiscible fluids, i.e., they absorb at the fluid interface covering the maximum fraction of the area available to minimize the contact line between the two immiscible fluids, reducing the interfacial tension [29]. This leads to the microgel flattening at the interface in such a way that it maximizes the occupied area, which can push the microgel to adopt a conformation at the interface characterized by a larger diameter (more than twice the diameter) than that of a bulk microgel. This is the result of the competition between the internal elasticity of the polymeric network and the interfacial tension, which favors the anisotropic deformation of microgels adsorbed at the fluid interface [9,22,28,30,31], leading to an energetic landscape characterized by high values of the adsorption energy and strong inter-particle capillary interactions [32,33,34,35,36,37]. However, in addition to this, the region of the microgel immersed in the aqueous phase remains solvated and swollen at temperatures below the characteristic VPT, adopting the so-called “fried-egg” conformation [38,39]. A sketch of the conceivable conformational change of microgels upon their adsorption to a fluid interface is shown in Figure 1a.

The spreading of microgels at the interface is strongly limited by the network elasticity of the particles, which is commonly determined by the cross-linking density of the particles, and the length of the chains between two cross-linking points associated with the polymer sub-chain elasticity. Consequently, the maximum deformation of microgels adsorbed at a fluid interface results from an intricate balance between the decrease in the interfacial energy and the elastocapillary length of the microgel (*L*_EC_ = *γ_S_*/*E*), which evaluates the ratio between the interfacial tension of the solid *γ_S_* and its Young’s modulus *E* [22].

The cross-linking density has an impact on the interfacial tension decrease observed when microgels are adsorbed at a fluid interface. In particular, the higher the cross-linking density, the larger the interfacial tension decrease and the weaker the microgel flattening at the interface [40]. Furthermore, the microgel size and its deformability play a very important role in the equilibrium shape of the microgel at the fluid interface that, in turn, has been observed to play a key role in the stabilization of the interface. On one hand, small microgels, with a radius below the *L*_EC_, undergo a strong deformation of their shape at the interface, which leads to the formation of liquid-like films of oblate-like microgel disks. On the other hand, large microgels (with a radius above the value of *L*_EC_) undergo an almost negligible change in their shape upon adsorption at the fluid interface, which results in behavior reminiscent of that expected for non-deformable colloids [40].

**Figure 1 polymers-14-01133-f001:**
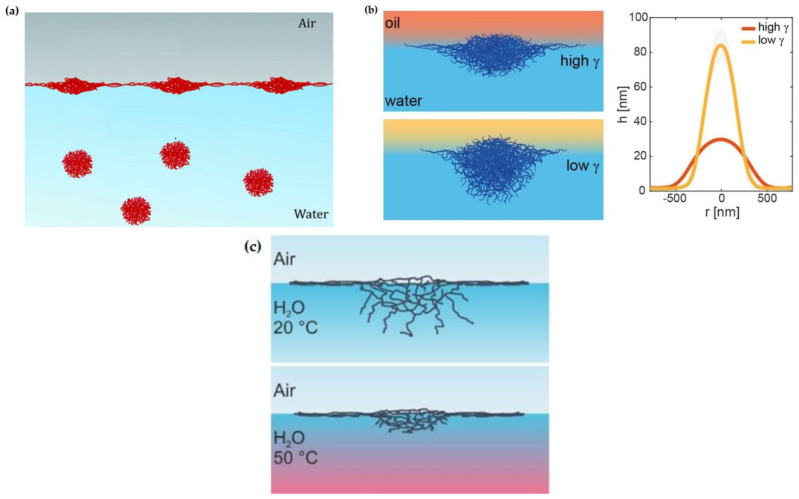
(**a**) Sketch of the microgel deformation upon adsorption to a fluid interface. Adapted from Deshmukh et al. [28] with permission from the Royal Chemical Society, Copyright (2014). (**b**) Conformation and dimension of microgel particles (characterized by the microgel radius at the contact length, *r*, and height, *h*) at water/oil interfaces with different interfacial tensions (*γ*). Reprinted from Vialetto et al. [41] under the Creative Common Attribution License 4.0, 2022. (**c**) Sketch of the conformation of the PNIPAM microgel at a water/vapor interface at 25 °C and 50 °C. Reprinted from Harrer et al. [42] with permission from the American Chemical Society, Copyright (2019).

The intrinsic characteristics of the fluid interface can also control the equilibrium conformation of the microgel at the interface. Adsorbed microgels protrude into the two fluid phases according to the solvency of the fluids for the polymer chains yielding an asymmetric partitioning of the microgel across the interface (commonly denoted as 3D organization) [17,22,36,38,41,43,44,45,46,47,48,49,50]. In addition, the conformation of microgels adsorbed at a fluid interface can be tuned by varying the chemical nature of the fluid phases, i.e., the interfacial tension. Thus, the lower the water/oil interfacial tension for similar polymer solubility on the oil, the smaller the deformation of the microgel core [41]. Figure 1b displays the conformation and dimensions (characterized by the microgel’s radius at the contact length, *r*, and height, *h*) of the microgel particles adsorbed at the water/oil interface with different interfacial tensions, in which the solubility of the microgel particles remains similar. These results, obtained by atomic force microscopy, AFM, show that the higher the surface water/oil interfacial tension, the stronger the collapse of the microgels at the interface, i.e., microgel spreading at the interface is smaller and their 3D height is lower.

The above discussion is in agreement with the results obtained by Zhang and Pelton [51], who demonstrated that the interfacial tension decrease associated with the adsorption of microgels at a water/vapor interface was correlated to their spreading at the interface. For instance, the adsorption of thermoresponsive PNIPAM microgel particles decreases the interfacial tension of the water/vapor interface down to a value of about 43 mN/m at 25 °C. This value is only slightly modified by the increase in the temperature of up to 40 °C, or the change of the degree of microgel cross-linking. However, the latter parameter impacts strongly on the equilibration of the microgel-laden interface. Thus, the different deformation of the particles in bulk determines the differences in their conformation upon adsorption at the interface. In fact, particles undergoing strong deformations upon adsorption at the interface (low cross-linking density) increase the interfacial coverage quicker than less deformed (high cross-linking density) ones, achieving the interfacial saturation in shorter time scales. This may be interpreted considering that the lower bulk elastic moduli of microgels with a low cross-linking density favors their spreading at the interface. However, the steric hindrance associated with the external coronae reduces the spreading rate of microgels at the interface as the equilibrium state is approached [23]. On the other hand, the absence of significant changes of the interfacial tension of microgel-laden interfaces upon temperature changes can only be understood considering that the interfacial confinement hinders the VPT [38,52]. This is the result of the counteractive role of the intrinsic bulk elasticity of the microgels and the interfacial tension on the microgel conformation, which alters the free energy balance governing the VPT [42]. Thus, the heating of PNIPAM at a water/vapor interface results in the collapse of the core immersed into the aqueous phase, whereas the corona attached to the interface remains unaltered. This makes that microgels adsorbed at the fluid interface present a “fried-egg” or core-corona conformation, independent of the subphase temperature. Figure 1c displays a sketch showing the subtle changes to the conformation of adsorbed microgels upon the change in the subphase temperature.

Microgel adsorption at temperatures below the volume phase transition occurs in a swollen conformation, with the microgel trying to cover the maximum possible area [35,36,53]. The situation becomes very different when microgels adsorb at temperatures above the volume phase transition. In this condition, microgels adopt a collapsed conformation, requiring higher microgel concentrations for ensuring an effecting coating of the fluid interface. This results in a dense packing of the microgel-laden interface and jamming phenomena [54,55,56]. A schematic representation of the different conformations of the microgels at the interface upon adsorption above and below the volume phase transition temperature (VPTT) is shown in Figure 2.

The interfacial coverage also plays a very important role in the control of the deformation of microgel particles upon their adsorption at fluid interfaces. At a low interfacial density, particles adsorb at the interface with a stretched-out conformation to ensure the maximization of the occupied area. This results in a favorable entropy contribution, which allows the energy penalty associated with the elastic deformation of the microgel particle to be overcome, leading to microgel-laden fluid interfaces where the particles are mostly immersed in the polar phase [57]. This conformation of the particles drives the formation of liquid-like monolayers or clusters at the interface assembled through capillary interactions [47,58,59]. The increase in the interfacial coverage reduces the inter-particle distance, which drives the collapse of the particle at the interface [11], leading to the crystallization of the microgels in hexagonal arrays characterized by close contact between the coronas of adjacent particles, i.e., the increase in the interfacial coverage drives the formation of films with a solid-like hexagonal packing [60].

## 3. Inter-Microgel Interactions at Fluid Interfaces

The polymer/colloid duality of microgels leads to a very different energetic landscape than that found for hard particles at fluid interfaces [11]. In particular, the Van der Waals interactions, which present a central role for hard particles adsorbed at fluid interfaces, are almost negligible when soft particles are considered. Furthermore, the possibility of undergoing shape deformation introduces new degrees of freedom in microgels at fluid interfaces [35,48,57,61,62]. In particular, the existence of embedding and deformation modes in soft particles influences the inter-particle interaction potential, which is dictated by a complex interplay of interactions, mainly dominated by capillary and steric ones [63,64]. Furthermore, electrostatic interactions in the microgel-laden interface are very different to that found for hard particles, due to the complex interplay between electrostatic effects, swelling, and collective mechanical properties [17]. Interestingly, Geisel et al. reported that electrostatic interactions play an almost negligible role in the control of the microgel deformability [59].

Last, but not least, strong capillary interactions arise between microgels adsorbed at a fluid interface, which may be understood considering the long contact line of adsorbed microgels, as well as their roughness and heterogeneity [32,65]. The important contribution of the capillary interactions was evidenced by Cohin et al. [66], who showed the ability of microgels to cluster upon adsorption at fluid interfaces. This was attributed to the existence of long-range capillary interactions between the particles. For instance, capillary forces (quadrupolar attractive interactions) govern the aggregation of big microgels at the water/oil interface for almost negligible surface pressure values [49,63]. A similar behavior was also reported for microgels at water/vapor interfaces [67]. The corona that induces a steric stabilization of the microgels in bulk yielded very strong capillary interactions between the microgels, resulting in their aggregation even at very low values of the interfacial coverage [68]. Thus, at a low interfacial coverage, the interactions are dominated by the highly stretched corona, independent of the temperature. However, at a high interfacial coverage, the role of the microgel portion inside the aqueous subphase is prevalent and controls the inter-microgel interactions [45].

## 4. In-Plane Organization of Microgels at Fluid/Fluid Interfaces

The ability of microgels to be distorted or collapse under compression leads to a more complex assembly at interfaces, in comparison to the incompressible hard colloidal particles [69]. In particular, the core-corona morphology of microgels defines its organization at fluid interfaces, and hence their phase behavior [17,45].

Bockenek et al. [45] defined, for temperatures below the volume phase transition, the existence of up to five different regimes in the phase behavior of microgels adsorbed at fluid interfaces as a function of the interfacial coverage, independent of their specific chemistry. These regimes are identified as (i) the diluted state, (ii) corona-to-corona contact, (iii) isostructural phase transition region, (iv) core-to-core contact, and (v) monolayer failure. However, the core-to-core contact regime disappears at temperatures above the VPTT. The existence of the regime corresponding to the corona-to-corona contact is surprising and can only be explained in terms of the swelling thermodynamics. Thus, considering the size of a neutral microgel in bulk, its size is determined by a complex interplay between the free energy and the temperature-dependent solvent–polymer mixing free energy [70]. This framework needs to be extended when microgels are adsorbed at a fluid interface to include an additional term accounting for the surface free energy gain, which is greater than the mixing ones, and nearly constant with the temperature [9,22,38,42,71]. This leads to a situation in which the part of the microgel network in contact with the interface remains stretched until the elastic contribution can counteract the surface free energy contribution. The thermoresponsiveness of the portion of the microgels protruding into the aqueous phase remains unaltered. Figure 3 shows the change in the surface pressure, *Π*, as a function of the number of microgels per area unit, *N*_AREA_, at two different temperatures, and the corresponding AFM images obtained for the different states of the monolayer. The transition between the diluted states to the regime characterized by the corona-to-corona contacts is accompanied by a sharp increase in the surface pressure. A further increase in the surface coverage pushes the system to the emergence of an isostructural phase transition. This is characterized by the melting of the initial lattice, and the formation of a second hexagonal phase with a smaller lattice parameter; during this transition, the surface pressure changes very slowly with the increase in the interfacial coverage. Once the second hexagonal phase is completely formed, the behavior becomes very different depending on the temperature. At high temperatures (above the volume phase transition temperature), the monolayer fails, probably due to the buckling, formation of multilayers, or desorption [58,72]. On the other hand, at temperatures below the volume phase transition, an additional phase characterized by a further increase in the surface pressure emerges before the monolayer failure is observed (regime iv). However, the monolayer failure occurs at similar values of the surface pressure independent of the temperature. This can be understood considering that there is a maximum amount of polymer segments that can be confined at the interface before the failure. This limit can be reached by two different routes: (i) the increase in the lateral packing of the film, forcing a 2D collapse, or (ii) the increase in the temperature, forcing a 3D collapse.

The intrinsic structure of microgel particles influences the phase transitions observed in interfacial films. The large corona of microgels with a low cross-linking density offers strong resistance to the phase transitions concerning the situation found for microgels with a high cross-linking degree. Therefore, the specific molecular architecture of the microgel particles allows controlling the organization of the particles at the fluid interfaces, with the dimensions of the microgel corona dominating the assembly by introducing an energy barrier, which makes the collapse difficult [69]. Furthermore, the lateral organization of microgels at fluid interfaces emerges strongly dependent on the average size of the particles, as demonstrated by Scheidegger et al. [49]. They studied the interfacial organization of microgels with 2 different diameters (210 nm and 1.45 μm) at a water/oil interface, and found very different behaviors, with the diverse phases emerging as the interfacial particle density was increased. The largest colloids present behavior with reminiscences of what is expected for core–shell particles with a hard core and a deformable external shell. This leads to a capillary-driven clustering at a reduced interfacial density, followed by a solid–solid transition when the compression degree is increased. The solid–solid phase transition occurs between two phases with a hexagonal crystalline order, but with differences in their lattice constants corresponding to the different origin of the phases. Thus, the initial phase emerging from the shell–shell inter-particle contacts is transformed in the final crystalline phase in which the ordering is governed by core–core contacts. This results from the nucleation of clusters of particles in core–core contacts, which drives the melting of the surrounding shell–shell crystal until the final interfacial structure governed by the second phase is obtained [72]. On the contrary, the smallest colloids do not undergo any aggregation, showing a smooth transition from a hexagonal lattice to a dense disordered monolayer. Geisel et al. [73] showed that both the decrease in the cross-linking density and the absence of the core in microgel particles are critical parameters that hinder the formation of ordered hexagonal arrays at high values of the interfacial coverage. This is the result of their high deformability, which facilitates the contacts between microgels at the interface, even from the relatively low values of the interfacial coverage.

Further insights into the organization of microgel particles adsorbed at fluid/fluid interfaces can be obtained by using computational approaches, e.g., dissipative particle dynamic simulations [74]. This provides information related to the importance of the cross-linking density of the microgels, and their compatibility with the fluid phases for controlling the deformation of the particles at the interface. Thus, the increase in the interfacial density reduces the spreading of the microgels at the interface, passing gradually from an oblate-like conformation (diluted systems) to a quasi-spherical one at the highest values of the interfacial density. Therefore, the increase in the interfacial density drives the thickening of the microgel monolayer, reducing its roughness. Conversely, the shrinking of the microgels adsorbed at a fluid interface as a result of the worsening of the solvent quality, leads to the decrease in the monolayer thickness and the rupture of the monolayer as a result of the increase in the unfavorable contacts between the two fluid phases. The effect of the cross-linking density of microgels on their in-plane organization at fluid interfaces was furtherly explored by Picard et al. [75]. They reported that PNIPAM microgels with a low cross-linking density self-assemble at the water/vapor interface, forming a hexagonal phase characterized by single lattice parameters undergoing a transition to more disordered phases, with the increase in the interfacial coverage. This is the result of a conformational rearrangement of the microgel particles from a highly flattened conformation with many polymer segments directly adsorbed to the interface, to an entangled flattened formation resulting in compact monolayers at the highest values of the interfacial coverage. The increase in the cross-linking density of the microgel particles changes the picture, resulting in the emergence of a coexistence between the two phases with a hexagonal order characterized by two different lattice parameters. Figure 4 shows the different in-plane-organization of microgels with different cross-linking densities as a function of the interfacial coverage.

The electrostatic interactions are also very important for the control of the in-plane organization of microgel particles at fluid interfaces. This can be understood as a consequence of the different deformability of the charged microgel particles. Thus, charged microgels at water/fluid interfaces undergo a higher compression than uncharged ones at low values of the interfacial coverage, and, consequently, the area occupied by the particles is reduced with the reduction in the charge density. This results from the impact of the inter-particle interactions in the lateral organization of the particles at the interface. However, the effect of the charges in the microgel compressibility becomes negligible for large microgels, and hence the lateral inter-particle interactions do not modify the in-plane organization of microgel particles at the interface. This can be explained by considering the dominant role of the volume contribution to the free energy over the interfacial ones. The situation becomes very different, independent of the microgel size, when the interfacial coverage is increased. In this situation, the differences in the compressibility between charged and uncharged microgels lead to changes in the length-scale of the interactions occurring between the microgel fractions protruding into the aqueous phase for charged and uncharged particles, as a result of the different swelling degrees of the particles. Therefore, the increase in the interfacial coverage introduces the contribution of the out-of-plane interactions between microgel particles, which modify the in-plane organization of the particles at the interface [76]. The differences in the organization of microgels with different charge densities at the interfaces can be more clearly understood by comparing the surface pressure–area isotherms depicted in Figure 5, together with the conformation expected for the particles at the fluid interface characterized for the diameter of the projection of the particle at the interface (*D*_interface_).

The phase behavior of microgels at fluid interfaces can be modulated by the modification of some physico-chemical parameters affecting the microgel bulk conformation, such as the pH, temperature, or ionic strength. Thus, considering thermoresponsive microgels, such as PNIPAM microgels, the phase behavior emerges as rather independent of the temperature for low interfacial densities, which may be explained by considering the dominance of the highly stretched corona on microgel organization. This independence of the temperature disappears with the increase in the interfacial density, with the segments of the microgels immersed in the aqueous subphase starting to be relevant, governing the inter-particle interactions. These portions can undergo collapse, altering the isostructural phase transition. The above discussion points out the important role of the bulk stimuli-responsiveness of microgels to control their interfacial compressibility [69]. Therefore, even though the volume phase transition of PNIPAM microgels is commonly identified as an intrinsic molecular property, it is strongly modified upon interfacial confinement [42].

## 5. Response of Microgel Layers to Mechanical Perturbations

The response of soft particles adsorbed at fluid interfaces to mechanical perturbations emerges as strongly dependent on their deformability [20]. In particular, the ability of the external corona for undergoing rearrangements plays an essential role in controlling the relaxation mechanisms of microgels against mechanical stresses [34]. These depend on both Young’s modulus and the wetting of the microgels at the interface [9]. Furthermore, the rheological response of microgels adsorbed at a fluid interface depends on the particle concentration, but also on the inter-particle distance and lateral organization of the microgels at the interface. In fact, it is possible to identify up to four different regimes for the rheological response of microgel-laden fluid interfaces against dilation, depending on the interfacial coverage. At a very low interfacial coverage, the microgel layer mainly behaves as an elastic film of soft disks interconnected to networks as a result of the capillary interactions. The increase in the lateral packing above the interfacial coverage corresponding to a dense random packing for the microgel shells leads to a linear increase in the elastic modulus, which, in turn, can be considered as a reminiscence of the compression of soft disks. The onset of the third regime is characterized by the decrease in the storage modulus, which can be the result of the expulsion of the collapsed polymer layer from the interface. A further increase in the lateral packing pushes the systems to a situation in which the distance between the microgels becomes smaller than that corresponding to their hydrodynamic diameter in water. This leads to an increase in the elastic modulus as a result of the overlapping of the microgel cores [77]. In general, the adsorption of the microgels to the fluid interface significantly reduces the mobility at the interface, which is the result of the higher elasticity of the microgel-laden interface. This elasticity is enhanced as the microgel size decreases due to the different ability of rearranging at the interface. The smaller the microgel size, the faster its interfacial rearrangement [78].

Vialetto et al. [21] demonstrated that the control of the microgel structure during the synthesis process is a very interesting strategy for modulating the properties of the monolayers at fluid interfaces, providing a link between the particle conformation and the material properties. At small values of the interfacial density, it emerges that a mechanical response governed by the polymer corona formed within the interfacial plane by the most external chains, with the specific characteristics of the internal core playing a negligible role. However, the situation changes as the interfacial coverage increases, and the core starts to play a very important role in the control of the interfacial mechanics. The absence of the core favors the particle deformation in the direction orthogonal to the plane of the interface, which allows for continuous compression without any modification to the interfacial packing. This makes it possible that the monolayer can maintain the hexatic order, even at the highest values of the interfacial density. This situation is very different concerning core–shell particles, where the hexatic order drops with the increase in the interfacial density to drive the formation of disordered monolayers. It should be noted that the above behavior differs from that found for microgels in bulk, and hence it may be expected to be a very important contribution of the interfacial tension to the control of the mechanics of microgels. Thus, the polymer shell, providing stability against aggregation in bulk, can undergo deformations upon confinement at the fluid interface. This induces strong capillary interactions between microgels, which drives the particle aggregation at the interface [68].

The cross-linking density is also expected to modify the mechanical response of microgels adsorbed at fluid interfaces due to the impact of such parameters on the microgel deformability. The elasticity limit of microgel-laden interfaces increases with the cross-linking density. This can be understood considering that microgels behave as hard colloidal particles rather than soft polymer chains. Simultaneously, the interfacial tension decreases earlier for the monolayers of loosely cross-linked microgels upon the increase in the surface coverage [73,74]. The mechanical behavior of microgel-laden fluid interfaces is strongly dependent on the conformation adopted by the particles assembled at the fluid interface [61]. Thus, layers formed by swollen microgels present a more elastic character than those containing collapsed ones. The collapse of microgel particles at the interface leads to a decrease in the elastic contribution of the dilational response and the increase in the viscosity. Finally, the response of microgel layers against shear also emerges as strongly dependent on the microgel conformation. Swollen microgels behave with reminiscences of the behavior expected for soft gel-like materials, which undergo an elastic response against shear stresses, whereas layers of collapsed microgels are easily broken under the application of shear stress due to their compact and brittle character [61]. The shear rate influences the degree of deformation of microgels at the interface [79]. In fact, the application of high shear rates to a microgel-laden interface forces the microgel flattening at the interface, whereas low shear rates force the organization of microgels in a more laterally compressed conformation.

## 6. Microgel-Laden Interfaces on the Stabilization of Emulsions

The ability of microgels to adsorb at oil/water interfaces, reducing their interfacial tension, plays a very important role in the stabilization of a new time of Pickering-like emulsions, the so-called Mickering emulsions [35,80,81,82].

The stimulus-responsiveness of microgel particles, and their ability to undergo conformation changes dependent on the surrounding environment, have been widely exploited for emulsion stabilization. The environment affects the stability of microgel-stabilized emulsions, and the application of specific stimuli can be used for triggering the destabilization of the obtained emulsions [36,53]. Ngai et al. [83] used the responsiveness to pH, ionic strength, and temperature of PNIPAM microgels modified with carboxylic groups for triggering the volume phase transition in the particles, demonstrating that the adsorption of the microgel particles to octanol droplets allows the stabilization of oil-in-water (o/w) emulsions against coalescence and Ostwald ripening in a way that emerged strongly dependent on particle hydrophobicity and its charging state. In particular, swollen microgels with a hydrophilic character can induce an effective stabilization of the oil droplets on the continuous aqueous phase. However, the collapse of microgel particles prevents emulsion stabilization, resulting in phase separation, except under conditions in which particles are highly charged (high pH). Thus, highly charged microgel particles can stabilize emulsions, even after undergoing a temperature-induced collapse, which proves the important role of the charge screening on the destabilization of the oil droplets. This offers the possibility of forcing the destabilization of microgel stabilized Pickering emulsions by the reduction in the solution pH, or by increasing the solution temperature or its ionic strength, which leads to an enhanced hydrophobicity of the microgel particles [84]. Figure 6 summarizes the different configurations and stabilities of emulsions stabilized by microgels under different conditions.

The role of the microgel deformability on the stabilization of dodecane-in-water was addressed by Destribats et al. [36]. They found that deformable microgels tend to form a 2D network of connected microgels, with a significant overlapping of the peripheral chains. On the hand, the increase in the cross-linking degree or the temperature leads to a destabilization of the emulsions as a result of the reduction in the particle deformability and the microgel overlapping. A similar modulation of the properties of the obtained emulsions can be obtained by changing the microgel dimensions at the interface taking advantage of the different abilities of microgels with different sizes for undergoing deformation at the fluid interface. Thus, the increase in the microgel size leads to the formation of heterogeneous interfacial layers, which favors the bridging between neighboring droplets, leading to the formation of flocculated emulsions. The opposite behavior appears when small microgels are used, which allows the formation of fluid emulsions characterized by the presence of well-dispersed droplets [85]. Figure 7 represents the effect of different parameters affecting the properties of emulsions on the organization of microgel-laden interfaces.

The different ability of microgels for coating the droplet surface on the characteristics of the emulsions was also explored by Destribats et al. [86], who studied the use of whey protein microgels for the stabilization of triglyceride-in-water. They found important correlations between the bulk properties of the microgels and those of the obtained emulsions. Thus, the use of uncharged microgels leads to the formation of emulsions producing large drops, whereas the use of charged particles results in the formation of emulsions containing small flocculated drops. These differences are correlated to the conformation of the microgel assemblies at the water/oil interface. Neutral particles or particles with a screened charged density by salt addition adsorb at the interface, forming at least a monolayer of highly aggregated particles, which limits the coalescence phenomena. However, charged microgels cannot form a complete monolayer on the droplet surface, and hence they present a very limited ability for ensuring the protection of individual droplets. However, this type of conformation favors the bridging of neighboring droplets, leading to flocculation and stabilization of emulsions containing sparsely covered droplets.

Schmidt et al. [35] explored the effect of the charge density on the microgel on the stabilization of emulsion by using microgels formed by PNIPAM with different fractions of methacrylic acid. They found that the stabilization of the emulsions does not depend directly on the electrostatic interactions, even though the presence of the charges is a key factor in emulsion stability. This can be understood considering that the presence of charges favors the stabilization of the emulsions, because they allow the swelling and deformation of the microgel at the water/oil interface to modulate. On the other hand, the internal microgel softness and porosity determine that the properties of microgel-stabilized emulsions can emerge as very different from those of the emulsions stabilized by a molecular surfactant or hard colloidal particles (Pickering emulsions). Therefore, the ability of microgel to undergo deformations at the water/oil interface is very important in the stabilization of emulsions. The latter is worsened when microgels are not easily deformable at the fluid interface.

Destribats et al. [79] also proposed that the organization of microgels at the water/oil interface may be tuned by changing the methodology used for the preparation of the microgel-laden interface. This decisively impacts the stability of microgel-stabilized emulsions. The use of methodologies relying on the application of different energies allows for the control of the water/oil interface deformation and the microgel packing density. Thus, the strong flattening of the microgels at the interface induced when emulsions are prepared by methodologies based on the application of high shear rate deformation limits the maximum coverage of the droplets, and favors their flocculation (bridged emulsions). On the other hand, the interfacial coverage increases as a result of the adsorption of laterally compressed microgels, which prevent droplet flocculation when low shear rates are applied for emulsion preparation. Figure 8 displays a sketch of the effect of the shear rate on the droplet dispersion within the continuous phase.

## 7. Outlook

To date, the behavior of soft particles at fluid interfaces has been less explored in comparison to hard, inorganic particles. Therefore, we consider that it is required to extend the understanding of the function–structure relation of microgels, and other types of soft particles, confined within fluid interfaces. Interestingly, these systems add further complexity to the interfacial behavior by allowing the modulation of their shape in response to the interaction with the environment. Thus, the new energetic landscape associated with the adsorption of microgels to fluid interfaces resulting from the complex interplay between microgel bulk elasticity and interfacial tension forces can be exploited for controlling the assembly of mesostructures. These offer many opportunities for modulating the rheological and dynamic behavior of the microgel-laden interface in a way that can help in the stabilization of emulsions. In summary, the microgel stimulus responsiveness offers many possibilities for opening new avenues for the fabrication of reconfigurable soft interface-based materials.

## Figures and Tables

**Figure 2 polymers-14-01133-f002:**
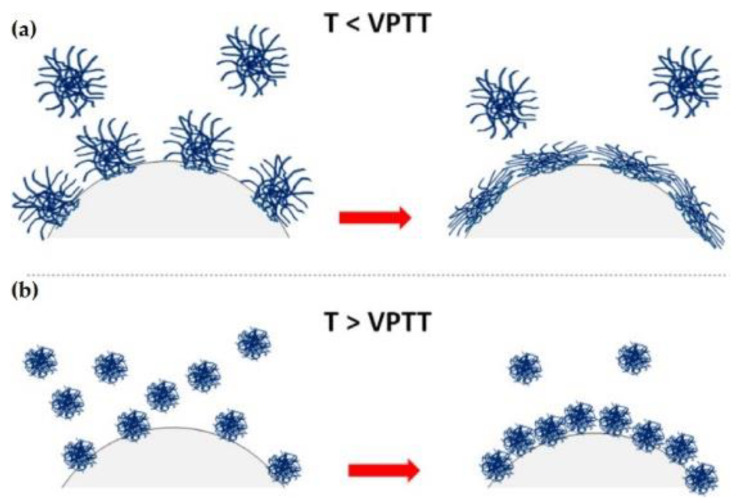
Sketch of the conformation of microgels adsorbed at a fluid interface from temperatures below (**a**) and above (**b**) the phase volume transition (VPTT: volume phase transition temperature). Adapted from Wu et al. [52] with permission from the American Chemical Society, Copyright (2019).

**Figure 3 polymers-14-01133-f003:**
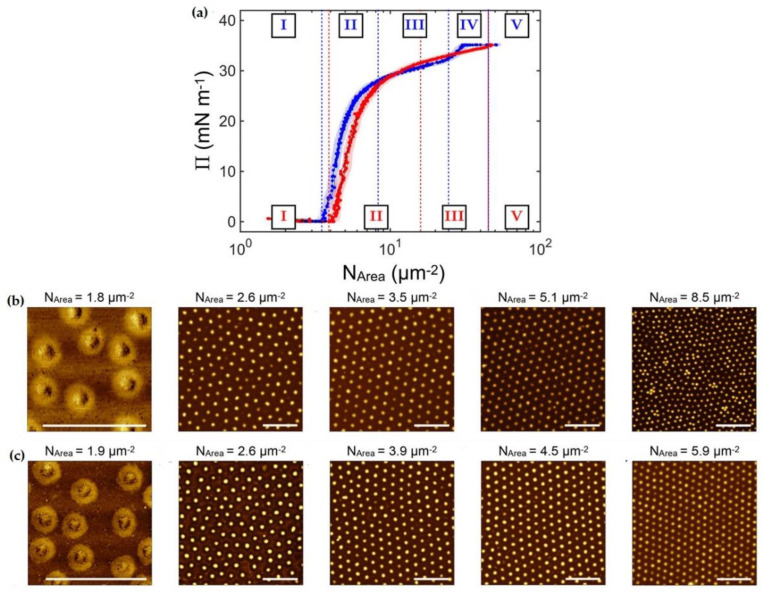
(**a**) Surface pressure (*П*) vs. number of microgels per area unit isotherms for PNIPAM microgels at a water/decane interface. The blue and red curves correspond to the isotherms obtained at 20 °C (below the volume phase transition temperature) and 40 °C (above the volume phase transition), respectively. The dashed lines indicate the onset of the different regimes, which are indicated by roman numerals in the corresponding colors. (**b**) Atomic Force Microscopy images of Langmuir–Blodgett deposits obtained for monolayers of PNIPAM microgels at a water/decane interface below the volume phase transition temperature (at 20 °C). (**c**) Atomic Force Microscopy images of Langmuir–Blodgett deposits obtained for monolayers of PNIPAM microgels at a water/decane interface above the volume phase transition temperature (at 40 °C). Adapted from Bochenek et al. [45] with permission from the American Chemical Society, Copyright (2019).

**Figure 4 polymers-14-01133-f004:**
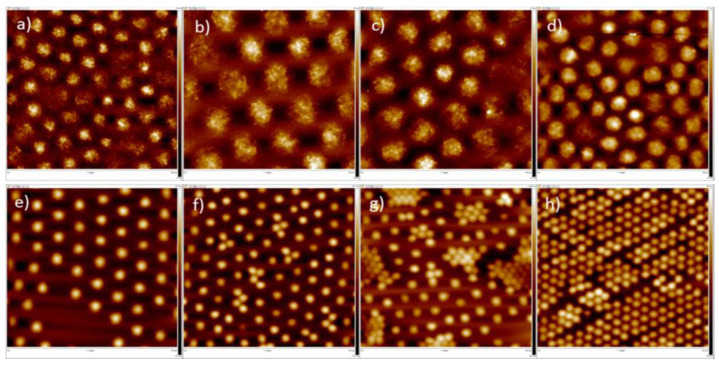
Atomic Force Microscopy images of Langmuir–Blodgett deposits obtained from low cross-linked (**a**–**d**) and high cross-linked (**e**–**h**) PNIPAM microgels at the water/vapor interface, obtained at increasing values of the interfacial coverage (from left to right). Reprinted from Picard et al. [75] with permission from the American Chemical Society, Copyright (2017).

**Figure 5 polymers-14-01133-f005:**
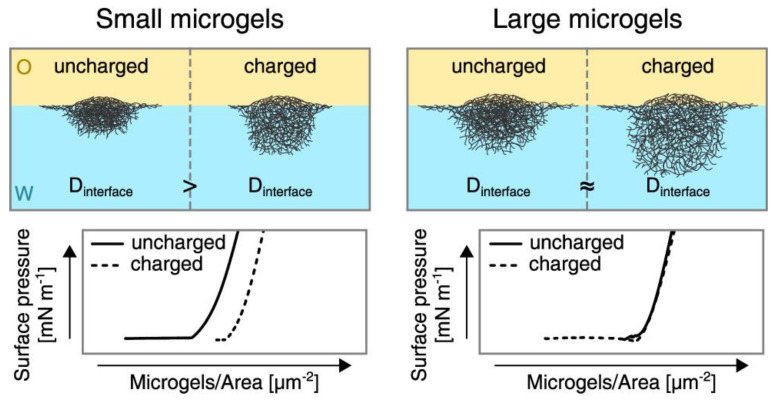
Effect of the size and charge of microgels adsorbed at a water/oil interface on the in-plane organization of the microgel film as a function of the diameter of the projection of the particle at the interface (*D*_interface_). Reproduced from Schmidt et al. [76] with permission from the Royal Society of Chemistry, Copyright (2020).

**Figure 6 polymers-14-01133-f006:**
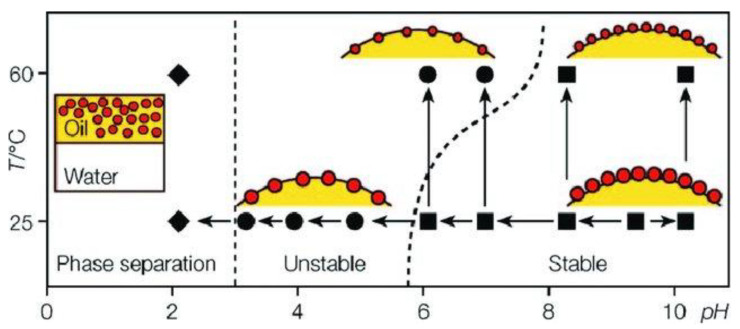
Stabilizing efficiency of PNIPAM microgel particles for oil-in-water Pickering emulsions as a function of pH and temperature. (■) Stable emulsions, (●) unstable emulsions and (◆) phase separation. Reproduced from Ngai et al. [84] with permission from the Royal Society of Chemistry, Copyright (2005).

**Figure 7 polymers-14-01133-f007:**
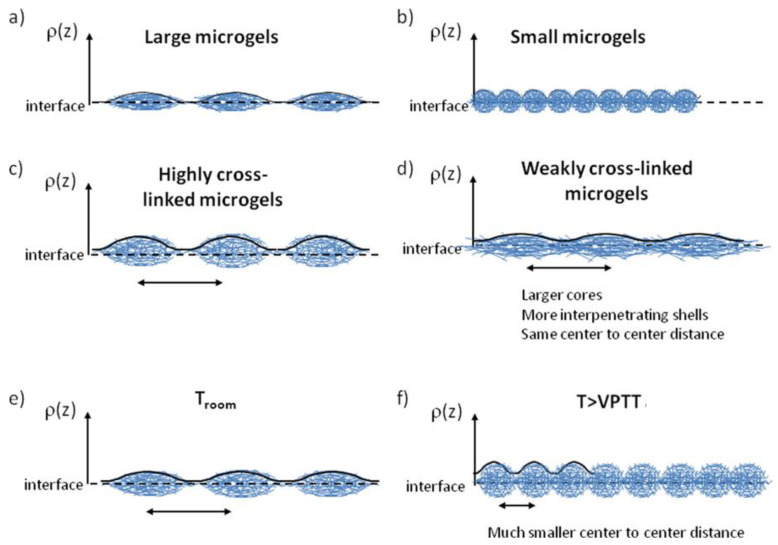
Effect of the different parameters affecting the emulsion properties on the organization of microgel-laden interfaces. (**a**,**b**) Microgel size. (**c**,**d**) Cross-linking density. (**e**,**f**) Processing temperature (VPTT: volume phase transition temperature). Adapted from Destribats et al. [85] with permission from the American Chemical Society, Copyright (2014).

**Figure 8 polymers-14-01133-f008:**
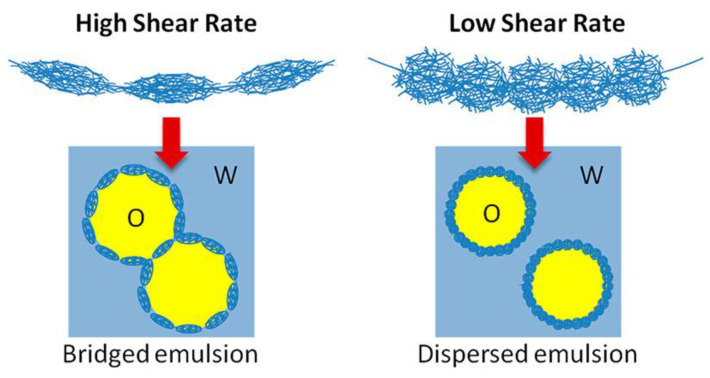
Effect of the shear rate on the dispersion of Mickering oil-in-water emulsions. Reprinted from Destribats et al. [79] with permission from the American Chemical Society, Copyright (2013).

## Data Availability

This work does not present any associated data.

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
