# Peer review of "Soft Colloidal Particles at Fluid Interfaces"

_polymers, 2022, doi:10.3390/polym14061133_

Round 1

Reviewer 1 Report

In general, this is a comprehensive and interesting review. Below are some questions and comments that need to be addressed.

  1. The abstract needs to be edited so it better reflects the motivation of this review article. Also, in line 12-13, instead of ‘report’, the authors should use ‘review’. No new results are being reported here, but a review on prior work in this field.
  2. Please increase the resolution of the figures when adapting from the other publications.
  3. Some discussion on how external shear deformation affect the emulsion structures could perhaps be moved to section 5: Response of microgel layers against mechanical perturbations.

Author Response

In general, this is a comprehensive and interesting review. Below are some questions and comments that need to be addressed.

  1. The abstract needs to be edited so it better reflects the motivation of this review article. Also, in line 12-13, instead of ‘report’, the authors should use ‘review’. No new results are being reported here, but a review on prior work in this field.

Following the recommendation of the reviewer, we have modified the abstract according to the comments.

  1. Please increase the resolution of the figures when adapting from the other publications.

The resolution of the figures has been improved when it has been possible.

  1. Some discussion on how external shear deformation affect the emulsion structures could perhaps be moved to section 5: Response of microgel layers against mechanical perturbations.

We have introduced the comment mentioned by the reviewer in Section 5.

Thank you for your comments, they were very useful for improving the manuscript.

Reviewer 2 Report

The review by Eduardo Guzmán and Armando Maestro is dedicated to the interesting topic, i.e., microgel behavior formed by cross-linked polymer networks. Their dual polymer/colloid character and the external stimulus responsiveness offer excellent opportunities to control the behavior of colloidal dispersions.

The authors did a great job of compiling a review of a significant number of works and made a thorough discussion of them. Undoubtedly, this manuscript is of great interest to many readers of Polymers journal. I have no critical remarks, and this review can be recommended for publication. Perhaps the authors should make a number of minor corrections to make the text clearer. All my comments are only advisory for the authors.

It would be nice to expand the description of the nature of microgels in the introduction. In particular, more details are needed on the causes of their dual nature, both in terms of their size and in terms of their chemical structure.

It would be good to repeat the decoding of physical quantities and abbreviations in the captions of the figures, which makes it possible to save time on searching for them in the text during a cursory or repeated reading.

So, in the caption to Fig. 1, it would be nice to remind the values of h and gamma. In the caption to Fig. 2 and 7, it would be good to explain the abbreviation VPTT again. In the caption to Fig. 3, it would be good to recall that Pi is surface pressure. And finally, in the caption to Fig. 5, it would be nice to recall what Dinterface is.

Author Response

The review by Eduardo Guzmán and Armando Maestro is dedicated to the interesting topic, i.e., microgel behavior formed by cross-linked polymer networks. Their dual polymer/colloid character and the external stimulus responsiveness offer excellent opportunities to control the behavior of colloidal dispersions.

The authors did a great job of compiling a review of a significant number of works and made a thorough discussion of them. Undoubtedly, this manuscript is of great interest to many readers of Polymers journal. I have no critical remarks, and this review can be recommended for publication. Perhaps the authors should make a number of minor corrections to make the text clearer. All my comments are only advisory for the authors.

It would be nice to expand the description of the nature of microgels in the introduction. In particular, more details are needed on the causes of their dual nature, both in terms of their size and in terms of their chemical structure.

Following the reviewer suggestion we have included a more detailed analysis of the microgel characteristics.

It would be good to repeat the decoding of physical quantities and abbreviations in the captions of the figures, which makes it possible to save time on searching for them in the text during a cursory or repeated reading. So, in the caption to Fig. 1, it would be nice to remind the values of h and gamma. In the caption to Fig. 2 and 7, it would be good to explain the abbreviation VPTT again. In the caption to Fig. 3, it would be good to recall that Pi is surface pressure. And finally, in the caption to Fig. 5, it would be nice to recall what Dinterface is.

We have added the information required in the Figure captions.

Thank you for your comments, they were very useful for improving the manuscript.